# Quantitative Groundwater Modelling under Data Scarcity: The Example of the Wadi El Bey Coastal Aquifer (Tunisia)



Hatem Baccouche [1], Manon Lincker [2], Hanene Akrout [1], Thuraya Mellah [1,3], Yves Armando [2] and Gerhard Schäfer [2,*]

1 Laboratory for Wastewater and Environment, Water Research and Technologies Center (CERTE) Technopark of Borj Cedria PB 273, Nabeul 8020, Tunisia; eng.hatem.baccouche@gmail.com (H.B.); hanene.akrout@gmail.com (H.A.); thouraya.mellah@esen.uma.tn (T.M.)
2 Institut Terre et Environnement de Strasbourg (ITES), UMR 7063 CNRS-Université de Strasbourg, 67084 Strasbourg, France; manon.lincker@live.com (M.L.); armando@unistra.fr (Y.A.)
3 Higher School of Digital Economy, University of Manouba, Manouba 2010, Tunisia
* Correspondence: schafer@unistra.fr

**Abstract:** The Grombalia aquifer constitutes a complex aquifer system formed by shallow, unconfined, semi-deep, and deep aquifers at different exploitation levels. In this study, we focused on the upper aquifer, the Wadi El Bey coastal aquifer. To assess natural aquifer recharge, we used a novel physiography-based method that uses soil texture-dependent potential infiltration coefficients and monthly rainfall data. The developed transient flow model was then applied to compute the temporal variation in the groundwater level in 34 observation wells from 1973 to 2020, taking into account the time series of spatially variable groundwater recharge, artificial groundwater recharge from 5 surface infiltration basins, pumping rates on 740 wells, and internal prescribed head cells to mimic water exchange between the wadis and aquifer. The quantified deviations in the computed hydraulic heads from measured water levels are acceptable because the database used to construct a scientifically sound and reliable groundwater model was limited. Further work is required to collect field data to quantitatively assess the local inflow and outflow rates between surface water and groundwater. The simulation of 12 climate scenarios highlighted a bi-structured north—south behaviour in the hydraulic heads: an increase in the north and a depletion in the south. A further increase in the pumping rate would, thus, be severe for the southern part of the Wadi El Bey aquifer.

**Keywords:** groundwater recharge; data uncertainty; numerical modelling; climate change scenarios; semi-arid region

## 1. Introduction

The management of water resources in coastal Mediterranean regions (MED) remains a serious issue. The demand for water is constantly increasing as a result of the growth of large megacities and the expansion of multisector economic activities [1,2]. This results in an important need for fresh water and the high exploitation of coastal aquifers, which can lead to saltwater intrusion [3]. This issue, combined with contaminated surface water that percolates towards the aquifer, and climate change, seem to be additional factors endangering the sustainability of these aquifers [4].

In this context, Sustain Coast (Sustainable Coastal Groundwater Management and Pollution Reduction through Innovative Governance in a Changing Climate) is a research project that involves researchers, local communities, water stakeholders, and policymakers to develop and test innovative governance to enhance the management and reduce pollution of MED coastal water resources. In this project's framework, four study sites around the Mediterranean Sea were selected for implementing the project activities, among which was the Wadi El Bey watershed in the Cap Bon region of Tunisia. At this site, the Grombalia aquifer constitutes a complex aquifer system formed by shallow, unconfined, semi-deep,

and deep aquifers at different exploitation levels. The interest of the present study is on the upper aquifer, the Wadi El Bey coastal aquifer. The Grombalia shallow unconfined aquifer has been under stress during recent decades. The overexploitation of renewable resources, which are estimated to be 51 million cubic meters annually, has increased to over 106 million cubic meters annually since 2015, leading to an overexploitation rate of about 208% [5], accompanied by water and soil salinization and alarming surface water pollution. However, it has been observed that the water table is rising in some areas. This may be because some wells are no longer in use, and pumping needs from groundwater are declining because more water is being transferred from the northern region of the country. The groundwater quality has significantly declined due to saltwater intrusion, increased nitrate contamination, and an increase in organic matter, especially near industrial areas. In addition, recurrent droughts have also had an impact on groundwater resources, particularly in the Cap Bon aquifers, including the Grombalia aquifer [6].

A strategic study of Tunisia's hydraulic system up to 2030 [6] has indicated that over the last 30 years, groundwater exploitation has increased from 710 million $m^3$ to 2133 million $m^3$, with the number of wells increasing significantly, and several aquifers are now overexploited. One of the objectives of the Sustainable Water Integrated Management and Horizon 2020 Support Mechanism (2016–2019) program [7], financially supported by the European Union, consists of creating an operational diagram of the Grombalia hydraulic system and finding solutions for better sustainable management of the region's water resources. In the final report, a series of technical recommendations, such as improving both the groundwater quality by identifying non-sustainable practices and surface water flow monitoring, were made. With the data available for this study, it was only possible to create a simple water balance for the Grombalia aquifer. In particular, a fully closed water balance would provide an overview of the measures required to maintain the aquifer in good condition. In addition, a more detailed analysis could be performed with a 3D groundwater model to support future scenarios and policy analysis.

Numerical modelling is one of the most relevant tools for investigating and understanding groundwater systems, estimating the effects of certain groundwater management methods, reconstructing the evolution of a hydrological system over time, and forecasting the effects of future actions or hydrological and climatic evolution. To achieve these results with numerical models, it is important to keep in mind that a model cannot represent the real world in every detail but is always a simplified representation of the complex reality of the natural system [8]. The level of detail of a model must be determined based on the purpose, the availability of data, and the limitations of the implementation process, as any unnecessary complication can be ineffective and time-consuming. One of the most important tasks in groundwater modelling is identifying the model area and its boundaries [9,10]. The criteria for selecting hydraulic boundary conditions are primarily related to topography, hydrology, and geology. Boundaries, such as impermeable areas and potentiometric surfaces controlled by surface water or inflow boundaries along rock formations, may be generated. To assess the path lines and residence times of pollutants, it is essential to use local-scale groundwater models under transient flow conditions [11,12]. Therefore, mathematical modelling of groundwater flow and water balances in alluvial aquifers requires time-dependent input data, such as water fluxes within the modelled region and across its boundaries [9,13].

To the best of our knowledge, Gaaloul et al.'s study [14] is the only scientific published work that addresses the numerical modelling of groundwater flow in the Wadi El Bey aquifer. They used the Visual MODFLOW tool to model the hydrodynamic conditions of the unconfined part of the Grombalia Cap Bon aquifer under steady and unsteady flow conditions from 1972 to 2010. Their 2D groundwater model was based on a finite-difference mesh with constant cell sizes of 200 m $\times$ 200 m, corresponding to 9000 cells. The simulated results, obtained under steady-state and transient conditions, showed satisfactory agreement between the observed and calculated groundwater levels. They then examined the potential effects of current and future groundwater use on the hydrodynamic conditions

of the aquifer system beyond the year 2010 to the year 2030. Three aquifer simulation scenarios were proposed based on different use management practices.

In addition to extending the groundwater flow model to three dimensions to take into account the vertical flow components of the unconfined aquifer, the need to improve the existing groundwater flow model of Gaaloul et al. [14] was identified mainly in four areas. These included (i) the evaluation of spatially distributed, transient groundwater recharge in a simplified manner; (ii) the implementation of time-dependent prescribed water heads along the six major wadis; (iii) the consideration of artificial groundwater recharge (between 1990 and 2015) from five surface infiltration basins; and (iv) the modelling of well-defined climate change scenarios to assess the spatiotemporal influence of future rainfall on groundwater reserves in the near future, midterm, and long term.

The main objective of the present numerical study was to construct an improved groundwater model using the interactive Finite Element subsurface FLOW system (FE-FLOW) to quantify the water budget of the shallow aquifer, simulate groundwater level changes under steady-state and transient conditions, and assess the influence of natural recharge on groundwater levels under different climate changes.

The complex hydrogeological setting of the investigated area needs to be understood. The present numerical results provide an example of modelling complex coastal aquifers with missing information of surface water levels and limited knowledge of spatially distributed vertical groundwater recharge.

In addition to being useful for Tunisia, this case study provides an illustration of a methodology that includes hydrological data in a groundwater model to gain insights into the water resources of a semi-arid region where data are lacking. In the present study, we used improved datasets for hydrological modelling and a physiography-based water balance to estimate aquifer recharge. Moreover, this approach may be applied to other groundwater systems.

## 2. Study Area

### 2.1. Characteristics

The Wadi El Bey site is located almost 40 km south of the Tunisian capital. It extends over an area of 391 km$^2$ and is home to approximately 24,500 inhabitants (Figure 1). It is bordered to the north by the Gulf of Tunis and the Tekelsa Hills, to the west by the Bou Choucha and the Halloufa mountains, to the south by the Hammamet Hills, and to the east by the Abderrahman Mountains and the oriental coastal highlands [15,16]. Surface flow occurs mainly in six wadis towards the north, reflecting the regional topographic gradients.

The Grombalia plain watershed is drained by six main wadis: Oued Saraya, Oued Defla, Oued Ejjorf, Oued El Bey, Oued Sidi Said, and Oued Bezirk. Around the Grombalia plain, the wadis steeply slope. The central wadi, Oued El Bey, discharges into the El Maleh Sebkha, which is close to the Mediterranean Sea. Only these six main wadis were included in our numerical study because they carry water year-round and do not dry out.

The study area is part of the Cap Bon peninsula (northeastern Tunisia). It corresponds to an ancient gulf that is wide open to the northwest and connects with the paleogulf of Tunis. Climatically, the region is a semi-arid zone and is characterized by an average annual rainfall of approximately 500 mm per year [17,18], an average annual temperature of 18 °C [19], relatively mild winters, and hot, dry summers. It should be noted that from 1972 to 2010, very high annual precipitation was observed in 1972 and 1982, with 724 mm and 819 mm, respectively [14]. Structurally, the area is a collapse trough linked to the subsidence of the upper basement and filled by quaternary deposits. The top of the quaternary series is generally sandy. Towards the centre of the Grombalia Basin, the quaternary strata are marneous, gypsiferous, and saliferous from a depth of 130 m [20].

Economic activities have resulted in a high degree of pollution in the Wadi El Bey watershed. Industry, agriculture, and insufficient wastewater treatment seem to be the primary causes of pollution. Moreover, due to surface water contamination, the quality of

groundwater is declining. A steady and significant decrease in the water table in the Wadi El Bey coastal aquifer remains a serious issue.

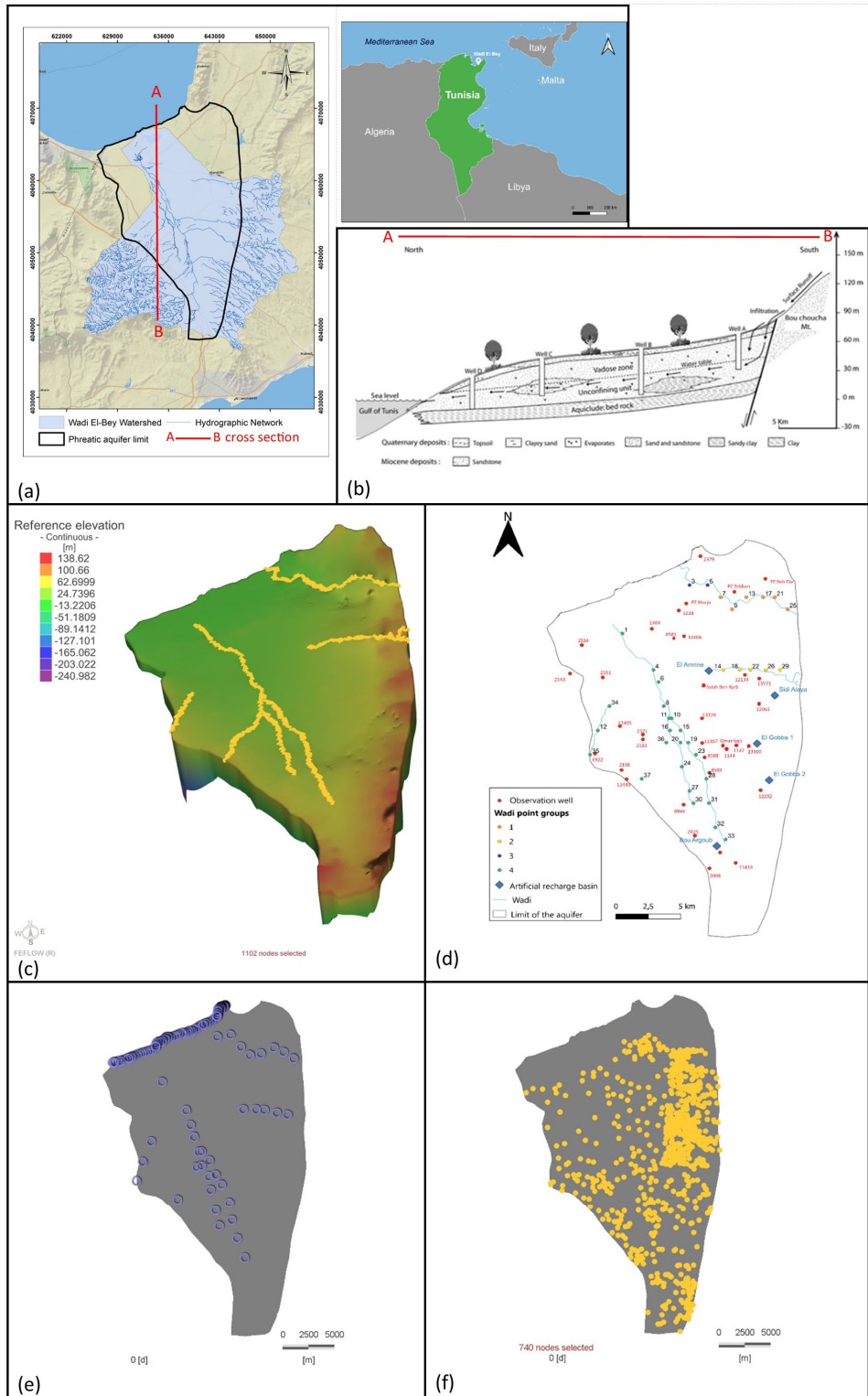

**Figure 1.** Location of the study site (**a**) and schematic view of the hydrogeological N—S cross section of the Grombalia unconfined aquifer [21] (**b**), 3D view of the numerical model with location of wadis (yellow dots) (**c**), location of observation wells, wadi points, and artificial recharge basins implemented in the groundwater flow model (**d**), and boundary conditions of the numerical model (plan view): prescribed water heads (blue circles) (**e**), and 740 pumping wells (yellow dots) (**f**).

*2.2. Hydrogeology*

In this study, we focused on the upper aquifer, the Wadi El Bey coastal aquifer (Figure 1a,b). The top elevation of the aquifer increases towards the south and southeast, reaching a maximum of 110 m in the area of Bou Argoub. The unconfined aquifer, with an average thickness of 50 m [22], is hosted in the quaternary continental sand, clayey sand, and sandstone deposits overlying 15 m thick clayey bedrock (Figure 1b). The hydraulic properties of the aquifer are characterized by significant lateral changes due to the heterogeneity of the lithological facies. The hydraulic conductivity values range from $5.4 \times 10^{-6}$ to $6.5 \times 10^{-3}$ m s$^{-1}$ [23]. The estimated specific yield ranges from 0.01 to 0.46. The flow direction of groundwater is generally directed towards the north. The groundwater levels monitored, for example, in 2010 varied between 0 and 70 m above sea level.

In 2015, the Grombalia aquifer was used by 8814 shallow wells with depths of less than 40 m, 6910 of which were equipped with motor-driven pumps. Additionally, 106 mm$^3$ of water was drawn annually, and 208% of renewable resources were exploited [5]. Notably, the volume of water extracted in 2015 and the number of active wells were three times greater than those in 1972. In comparison with the findings of Ennabli [20], the piezometric level of the Grombalia aquifer has generally decreased by approximately 10 m in the Belli region over the last 50 years. Thirty-four observation wells (Figure 1d) are located in the unconfined aquifer, where hydraulic heads have been regularly monitored at least once a year by the DGRE (Direction générale des ressouces en eaux) since 1973.

Despite artificial recharge operations between 1990 and 2015 via recharge basins (Figure 1d) and irrigation with water from the Mejerda–Cap Bon canal, the groundwater level of the unconfined aquifer continued to decline. The coordinates of the observation wells and artificial recharge basins can be found in the Supplementary Materials (Table S1).

## 3. Materials and Methods

### 3.1. Numerical Flow Model FEFLOW

FEFLOW is a modular 3D finite element groundwater flow model capable of simulating transient processes of flow and mass transport processes in subsurface water resources [24]. The general flow equation for saturated groundwater flow is formulated by applying the law of conservation of mass over a macroscopic control volume of porous media located in the flow field [25]. The net inflow into the volume must equal the rate at which water accumulates within the volume under investigation. Thus, the following expression can be obtained:

$$S_s \frac{\partial h}{\partial t} = \frac{\partial}{\partial x_i} \left( K_{ij} \frac{\partial h}{\partial x_j} \right) - Q \tag{1}$$

where the subscripts $i,j$ (= 1, 2, 3) represent the principal coordinate directions, $x$ represents the space coordinates [L], $t$ represents the time [T], $K$ represents the hydraulic conductivity [LT$^{-1}$], $h$ represents the hydraulic head [L], $S_s$ represents the specific storage [L$^{-1}$], and $Q$ represents the local sources and sinks per unit volume [T$^{-1}$].

The Galerkin finite element method was adopted to solve Equation (1) using automatic time step control via predictor–corrector schemes in the FEFLOW subroutines to perform the simulations. Initial and boundary conditions are required for the solution of the flow equation. The initial condition is the hydraulic head at the beginning of the simulation. The boundary conditions related to the primary variable h must be specified for the entire model boundary and may vary with time.

The dimensionless error criterion in FEFLOW was used for the iterations in the flow simulations and the automatic time-stepping process. The chosen error tolerance was $10^{-4}$ using the Euclidian L2 integral root mean square error (RMS) norm. To solve the symmetric matrix of the fluid-flow problem, we used the PARDISO director solver implemented in FEFLOW.

### 3.2. Boundary Conditions and Input Parameters of the Numerical Groundwater Model

The geometry of the aquifer was extracted from the map "Carte Agricole de Nabeul" from the Regional Commissary for Agricultural Development of Nabeul (CRDA Nabeul). The top layer of the numerical model was determined using the Shuttle Radar Topography Mission—Digital Elevation Model with a 30 m resolution. The bottom elevation of the phreatic aquifer was extracted from the lithostratigraphy of existing boreholes provided by the CRDA Nabeul.

FEFLOW applies a finite element technique, providing a discretized solution. For this, the modelling area is subdivided into nodes and elements as part of a mesh. In three-dimensional models, the elements are arranged in layers and the nodes in slices in the horizontal X–Y plane. A layer is bounded vertically at the top and bottom by one slice each. The constructed 3D numerical model was composed of three layers and four slices with 96,983 mesh elements and 195,116 nodes (Figure 1c).

The chosen boundary conditions for the flow are no-flow conditions along the eastern and western boundaries and prescribed head boundary conditions. The nodes where the hydraulic head is prescribed concern two groups of nodes, as shown in Figure 1e:

- Along the northern model boundary that corresponds to the Mediterranean Sea, the nodes of all four slices were implemented in the model with a constant hydraulic head equal to 0 m.
- Along the main wadis, 36 internal nodes (called wadi points) with fixed hydraulic heads were implemented to simulate water exchange between surface water and groundwater. The working hypothesis of introducing local nodes where the groundwater head is supposed to be equal to the water level in the wadi was motivated by the fact that no data regarding the monitored water levels in the wadis were available. In the steady-state flow model, the prescribed hydraulic heads of the chosen wadi points were extracted from the study of Gaaloul et al. [14] by digitizing these 36 wadi points located on their computed isolines of water heads. In the transient flow model, a time-dependent hydraulic head was introduced at each of the 36 wadi points. To do this, we applied the open-source geographic information system QGIS. In a first step, groundwater level maps were created for the months of April and October, from 1972 to 2019, based on the groundwater levels measured at the 34 observation wells using a triangular interpolation method. The groundwater level maps determined then formed the basis for extracting the time-dependent hydraulic heads of the wadi points [26]. Figure 2 shows the time-dependent hydraulic heads quantified for April and October at all the wadi points. The starting values in 1972 correspond to those extracted from Gaaloul et al. [14]. A strong temporal variation in the hydraulic head of up to 30 m was recorded for wadi points in Groups 1, 2, and 3 (Figure 2a–c), whereas a moderate variation of approximately 10 m was observed for the other wadi points. Furthermore, at two wadi points, WP 1 and WP 5, temporarily negative hydraulic heads were observed, which correspond to a groundwater level below sea level. This can be attributed to either the fact that both wadi points are close to the seashore or to a local artefact of the interpolation method in QGIS due to the sparsely distributed data.

Furthermore, a well-type condition was introduced in the form of a total of 740 wells (whose locations are officially known) (Figure 1f). In the steady-state flow model, a constant pumping rate of 155 $m^3 d^{-1}$ was applied to each well. The selected pumping rate was calculated on the basis of the total volume of water pumped in 1972, which was estimated by the DGRE at 41.86 million $m^3$ [5]. In the case of transient flow simulation, a time-dependent pumping rate was implemented (Section 4.2).

The spatial distributions of the hydraulic conductivity in the five main zones and the drainable porosity were adopted from the calibration results of Hammami [27] (Figure 3). The hydraulic conductivity and drainable porosity are between 4 and 80 $md^{-1}$ and between 0.2 and 0.4, respectively. As can be seen in the purple area, the spatial distribution of the drainable porosity deviates slightly from the five main zones of hydraulic conductivity.

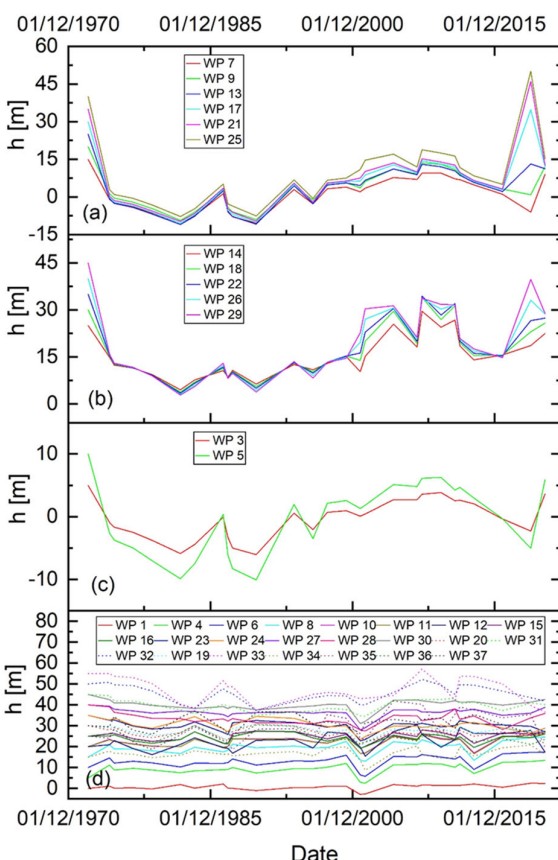

**Figure 2.** Hydraulic head variation evaluated for the 36 wadi points: (**a**) Group 1, (**b**) Group 2, (**c**) Group 3, and (**d**) Group 4. Notably, in the legend, the wadi points are abbreviated as WP.

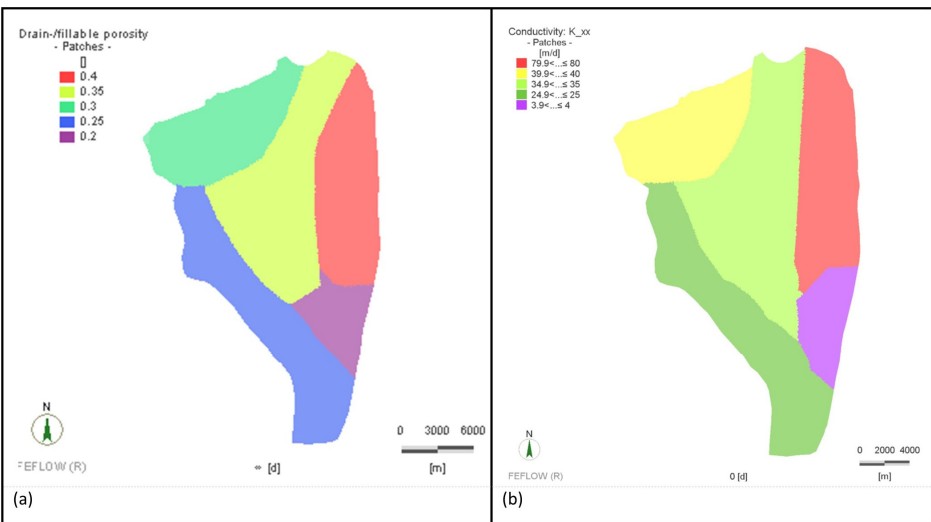

**Figure 3.** Spatial distribution of both drainable porosity (**a**) and isotropic hydraulic conductivity (**b**).

In the steady-state flow model, based on the mean monthly rainfall $R$ [LT$^{-1}$] and the potential infiltration coefficient *PIC*, which is a function of the soil texture, a spatially variable groundwater recharge *GR* [LT$^{-1}$] was applied. The *GR* was calculated for each areal zone $i$, considering the local values of both $R$ and *PIC* for each geological formation in the model domain [28]:

$$GR_i = R_i \times PIC_i. \tag{2}$$

The effective groundwater recharge amount used in the steady-state flow model decreased from north to south from 0.0012 to 0.0003 md$^{-1}$. Based on the five major zones

of hydraulic permeability (Figure 3), the mean annual rainfall of 724 mm (measured in 1972, [14]), and the soil texture-dependent *PIC*, we applied constant groundwater recharge to each of the five major groundwater recharge zones called 'RN Perm' (Figure 4a): 0.00089, 0.00029, 0.00119, 0.00119, and 0.00059 md$^{-1}$ to zone RN Perm 4, RN Perm 25, RN Perm 35, RN Perm 40, and RN Perm 80, respectively.

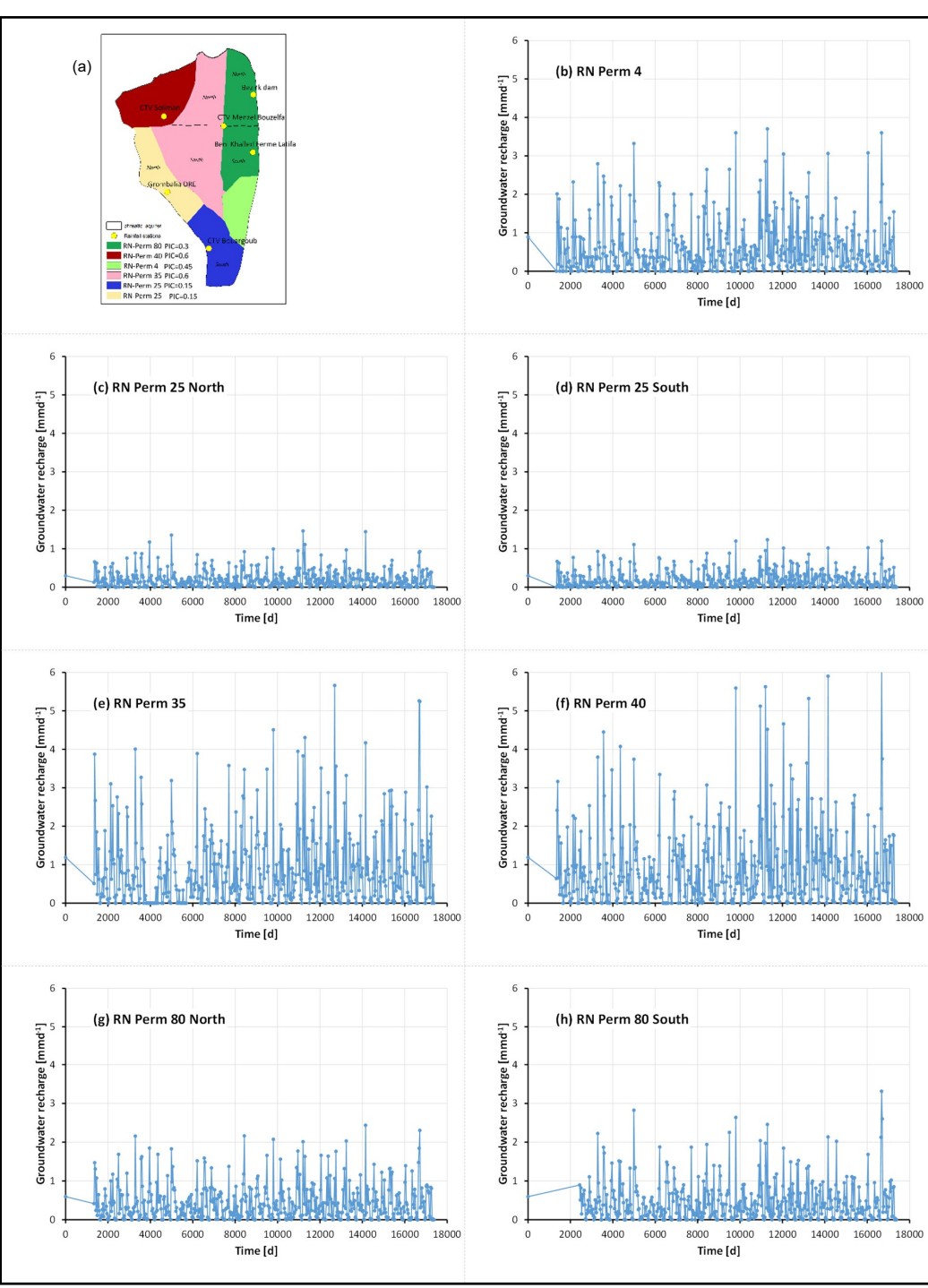

**Figure 4.** Groundwater recharge adopted in the transient flow model: (**a**) location of the selected seven zones and time-dependent (monthly variable) groundwater recharge for zones (**b**) RN Perm 4, (**c**) Perm 25 North, (**d**) Perm 25 South, (**e**) RN Perm 35, (**f**) RN Perm 40, (**g**) RN Perm 80 North, and (**h**) RN Perm 80 South. Note: *t* = 0 d and *t* = 17,503 d correspond to 1 January 1973 and 31 December 2020, respectively.

In the case of transient flow modelling, we used monthly variable rainfall data to assess the effective groundwater recharge as a function of time and space. Using Equation (2), daily rainfall data monitored at six gauging stations (Bezirk Dam, Beni Khalled Ferme Latifa, CTV Bou Argoub, CTV Soliman, CTV Menzel Bouzelfa, Grombalia DRE) combined with the soil texture-dependent *PIC* were adopted to quantify the spatially variable monthly groundwater recharge in 7 distinct zones. Rainfall data monitored at Bezirk Dam, Beni Khalled Ferme Latifa, CTV Bou Argoub, CTV Soliman, CTV Menzel Bouzelfa, and Grombalia DRE were applied to the zones 'RN-Perm 80 North (*PIC* = 0.3)', 'RN-Perm 80 South (*PIC* = 0.3)', 'RN-Perm 4 (*PIC* = 0.45) and RN-Perm 25 South (*PIC* = 0.15)', 'RN-Perm 40 (*PIC* = 0.6)', 'RN-Perm 35 (*PIC* = 0.6)', and 'RN-Perm 25 North (*PIC* = 0.15)', respectively. Groundwater recharge due to rainfall was applied to the first slice of the model aquifer. Figure 4 shows an overview of the locations of the seven selected zones and shows the time-dependent monthly groundwater recharge for each of the seven zones. The highest groundwater recharge was determined for the zones 'RN Perm 4', 'RN Perm 35' and 'RN Perm 40', which was due to the combination of high *PIC* and high monthly precipitation in the northern part of the study area compared to the southern part.

In addition, to take into account the contribution of deep aquifers by leakage, we applied a constant recharge rate of approximately $4.3 \times 10^{-5}$ md$^{-1}$ to the fourth slice of our model (which forms the aquifer bottom). This corresponds to an annual recharge from the deeper aquifer, estimated to be $7 \times 10^6$ m$^3$ [29].

*3.3. Climate Change Scenario Analysis*

The groundwater flow model was run for both historical input data and data from Regional Climate Models (RCMs) developed as part of the EURO-CORDEX project [30] in collaboration with the University of Parma.

The Regional Climate Models required a bias correction that was aimed at adjusting the raw outputs of the RCMs so that they better represented the statistical distribution of the data on the local scale. To do this, precipitation and temperature data observed over a 30-year control period were needed. The control period used for the bias correction spans from 1976 to 2005. To correct for systematic errors (bias) in the outputs of the regional climate models, we used precipitation data recorded during the period of 1976–2005 at the four weather stations (Bezirk Dam, Beni Khalled Ferme Latifa, CTV Bou Argoub, Grombalia DRE) located in the study area. The data showed no gaps in the time series. Since the temperature data recorded at the Nabeul weather station during the control period did not contain sufficient data to carry out bias correction, the gaps were filled with temperature data from the WATCH Forcing Dataset [31].

RCMs are advantageous for understanding the local climate of regions with complex topography. Historical data were used for downscaling and projecting into the future for two representative concentration pathway (RCP) scenarios: 4.5 and 8.5. The grid resolution of the RCMs used was 12.5 km (EUR-11 grid), and these RCMs were combinations of several general circulation models (GCMs) and RCMs. To obtain climate precipitation model data at the four gauging station locations considered, an inverse distance method (power of 2) was adopted that considered the nine cells closest to the specific station. After this interpolation, a bias correction was applied to adjust the raw outputs of the RCMs so that they better represented the statistical distribution of the observed data on a monthly scale. The climate model data were bias-corrected with reference to the 1976–2005 control period using the distribution mapping method [32] so that their cumulative distribution functions, at a monthly scale, agreed with those of the observed data in the chosen control period. The same correction estimated for the historical period was then applied for the future. More information and additional details on the downscaling and bias correction methods can be found in the studies of D'Oria et al. [33–36] and Secci et al. [37].

Daily precipitation data based on two representative concentration pathways, RCP 4.5 and RCP 8.5, were thus obtained at the four weather stations located inside the model area. The RCM simulation period spans from 1 January 1976 to 31 December 2098. Six

climate scenarios (out of 17 generated) were selected as the most important. The six climate scenarios were based on RCPs 4.5 and 8.5, resulting in 12 case studies. The regional climate models used for the climate change scenarios are listed in Table 1 in order of priority.

**Table 1.** List of RCMs used for climate change scenario analysis. Note: The selected RCMs are combinations of general circulation models and regional climate models; for example, RCM 'CNRM_CERFACS_CNRM_CM5_CCLM4_8_17' stands for the GCM 'CNRM_CERFACS_CNRM_CM5' combined with the RCM 'CCLM4_8_17'.

| Climate Scenario | Regional Climate Model (RCM) |
| --- | --- |
| 1 | 'CNRM_CERFACS_CNRM_CM5_CCLM4_8_17' |
| 4 | 'DMI_HIRHAM5_NorESM1-M' |
| 8 | 'ICHEC_EC_EARTH_HIRHAM5' |
| 9 | 'IPSL-INERIS_WRF381P_IPSL-CM5A-MR' |
| 12 | 'KNMI_CNRM-CM5' |
| 17 | 'MPI_M_MPI_ESM_LR_RCA4' |

Detailed information on the temporal variation in the monthly rainfall predicted at the weather stations of Bezirk Dam, Beni Khalled Ferme Latifa, CTV Bou Argoub, and Grombalia DRE by each of the six climate scenarios and for RCPs 4.5 and 8.5 can be found in the Supplementary Materials (Figures S1–S4).

## 4. Results and Discussion

### 4.1. Steady-State Flow Model (1972)

The hydraulic head distribution of the steady-state flow model computed for the reference year 1972 is shown in Figure 5 and compared to the results presented in Hammami's master's thesis [27] and reprocessed by Gaaloul et al. [14]. Our steady-state flow model seems to reproduce their findings correctly. The isolines of the hydraulic heads calculated at intervals of 5 m (Figure 5a) generally agree with the contour map (shown as black isolines) obtained from the field application. While the agreement in the northwestern part and in the central part of the aquifer is quite satisfactory, there are larger deviations in the southeastern part. Here, the isolines are represented by straight lines based on field observations, in contrast to the curved isolines in our numerical study. This part of the aquifer corresponds to the 'RN Perm 4' zone, which has the lowest hydraulic conductivity but a relatively high groundwater recharge, leading to a strong local increase in hydraulic head. In contrast, this effect was not observed in the study by Gaaloul et al. [14], who assumed constant groundwater recharge for the entire model area.

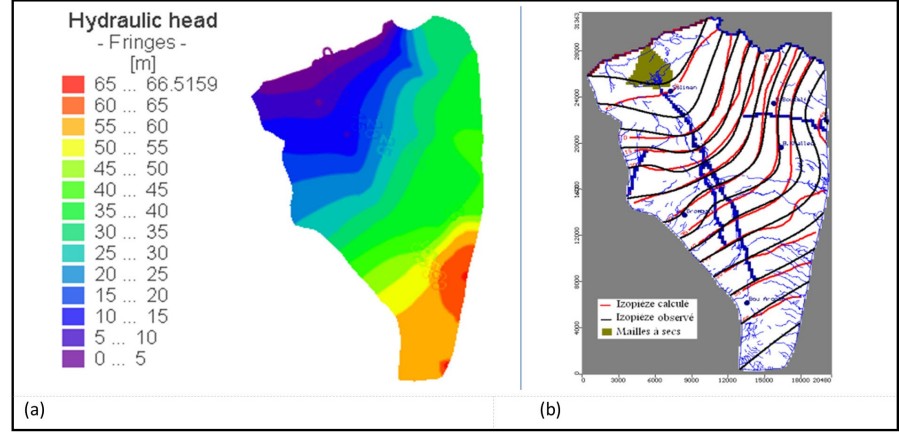

**Figure 5.** Hydraulic head distribution computed at steady-state flow conditions for 1972 (**a**) and compared to manually interpolated isolines obtained from field observations (dark lines) and isolines computed with MODFLOW (red lines [27]) (**b**).

By digitizing 70 virtual observation points located on their computed isolines of water heads, we compared our pointwise results to their "observed" hydraulic heads. The coordinates of the virtual observation points can be found in Table S1 of the Supplementary Materials. The distributions of the 70 virtual observation points are shown in Figure 6a. A detailed comparison is shown in the form of a scatterplot of the hydraulic heads computed by us with the results of Hammami [27] and Gaaloul et al. [14] (Figure 6b). The blue line represents the 1:1 line. Our model results are in good agreement with those of Gaaloul et al. [14], with a mean error ($\overline{E}$), a root mean square error (RMS), and a standard deviation ($\sigma$) of 3.57, 4.54, and 4.57 m, respectively. As mentioned above, the observed deviations are mainly due to the chosen type of groundwater recharge, which is the same in terms of annual water volume. However, the difference was caused by the uneven spatial distribution of groundwater recharge, which was deliberately chosen in our study.

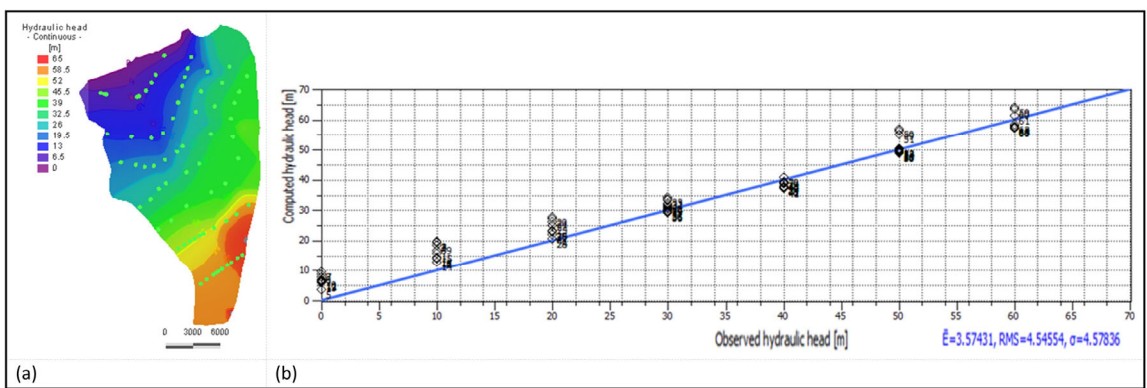

**Figure 6.** Locations of the 70 chosen virtual observation points (**a**) and cross-plot of our computed hydraulic heads against the "observed" hydraulic heads obtained in the previous numerical studies of Hammami [27] and Gaaloul et al. [14] (**b**). The blue line shown in Figure 6b represents the 1:1 line.

### 4.2. Transient Flow Model (1973–2020)

In our transient groundwater flow simulation for the period from 1973 to 2020, we used the hydraulic heads from the steady-state groundwater flow model as the initial groundwater levels. The transient boundary conditions include the time-dependent pumping rate, artificial aquifer recharge from five recharge basins (Figure 7), and transient groundwater recharge due to rainfall (Figure 4).

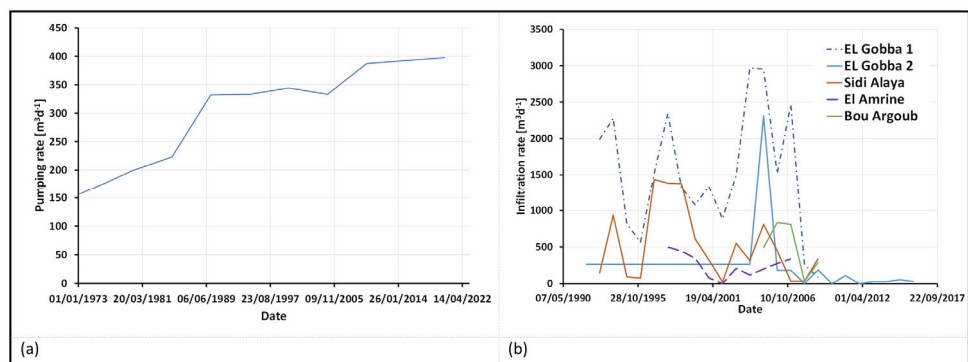

**Figure 7.** Pumping rate applied for each of the 740 wells from 1973 to 2020 (**a**) and infiltration rate implemented in the model for each of the five artificial recharge basins active between 1990 and 2015 (**b**).

As shown in Figure 7a, at each of the 740 wells (whose locations are officially known) (Figure 1f), we applied a time series of pumping rates ranging from 155 m$^3$d$^{-1}$ on 1 January 1973 to approximately 400 m$^3$d$^{-1}$ on 31 December 2020. In the period of 1985–1990, there

was a significant increase in exploitation of 49%. Between 1990 and 2000, exploitation of the Grombalia aquifer continued, but the increase was less significant than it was in the previous decade.

The artificial recharge of the aquifer from the five recharge basins was implemented in the numerical model as infiltration rates of water volume per day. As shown in Figure 7b, the greatest amounts of artificial recharge occurred between 1993 and 2008 at the El Gobba, Sidi Alaya, and El Amrine sites. From 1993 to 1997, no water infiltrated at the El Amrine site. The total volume infiltrated at the El Gobba site in 2004 was the highest in the period of 1993–2008 and exceeded 1 million m$^3$. The lowest quantity recorded in the period of 1993–2008 was 0.0094 million m$^3$, which was discharged at the Sidi Alaya site in 2001. The Bou Argoub infiltration basin was created in 2006 and operated for a very short period, with a maximum infiltration rate of approximately 800 m$^3$d$^{-1}$.

The hydraulic head distributions of the transient flow model computed for 31 December 1983, 31 December 1993, 31 December 2003, 31 December 2013, and 31 December 2020 are shown in Figure 8. In the contour maps of the hydraulic heads in Figure 8b–f, the 25 m contour line is highlighted as a reference of the mean groundwater level. This reference line shifts from north to south with increasing simulation time. This shows the global depletion of groundwater resources due to increasing water abstraction. The effect of local artificial recharge in the Sidi Alaya basin in the form of locally increased hydraulic heads is visible in the form of two highlighted green spots on the map created for 2003 (Figure 8d) in the northeastern part of the model area. Ten years later, the green area expanded further and occupied a large part of the aquifer (Figure 8e). The impact of the Bou Argoub infiltration basin is also visible in the contour map of 2013 (Figure 8e): in the southwestern part, a yellow spot in a green-dominated area indicates an increased hydraulic head due to the operation of the Bou Argoub recharge basin.

Table 2 summarizes the individual components of the water balance that were calculated using the transient flow model. Based on the quantified cumulative water volumes, the inflow and outflow rates for each component were determined by averaging over the entire simulation period and comparing with the inflow and outflow rates of the steady-state flow model. The effect of the artificial recharge of the aquifer from the five recharge basins is shown in Table 2 in the row 'Well inflow'. It should be noted that the groundwater recharge rate of the steady-state flow model is approximately 34% higher than the averaged groundwater recharge rates of the transient flow model. This is most likely due to the fact that the annual precipitation in the reference year (1972) was very high at 724 mm compared to the mean annual rainfall of 500 mm. As shown in Table 2, the averaged pumping rates of the transient flow model were almost twice as high as the pumping rates of the steady-state flow model.

**Table 2.** Water balance components of the transient flow model: cumulative water volume computed for the entire model domain as well as inflow and outflow rates averaged over the simulation period (1973–2020), which are compared with the flow rates of the steady-state flow model.

| | | Cumulative Water Volume (1973–2020) | Flow Rates Averaged over 1973–2020 | Flow Rates of the Steady-State Flow Model (1972) |
|---|---|---|---|---|
| | | ($\times 10^9$ m$^3$) | ($\times 10^5$ m$^3$d$^{-1}$) | ($\times 10^5$ m$^3$d$^{-1}$) |
| Fixed head boundary | inflow (+) | 2.65 | 1.51 | 0.85 |
| | outflow (−) | 4.21 | 2.40 | 2.96 |
| Wells | inflow (+) | 0.02 | 0.01 | - |
| | outflow (−) | 4.02 | 2.30 | 1.14 |
| Groundwater recharge (+) | | 4.24 | 2.42 | 3.24 |
| Storage | release (+) | 5.44 | 3.10 | - |
| | capture (−) | 4.15 | 2.37 | - |
| Imbalance | out (−) | 0.015 | 0.0085 | 0.0005 |

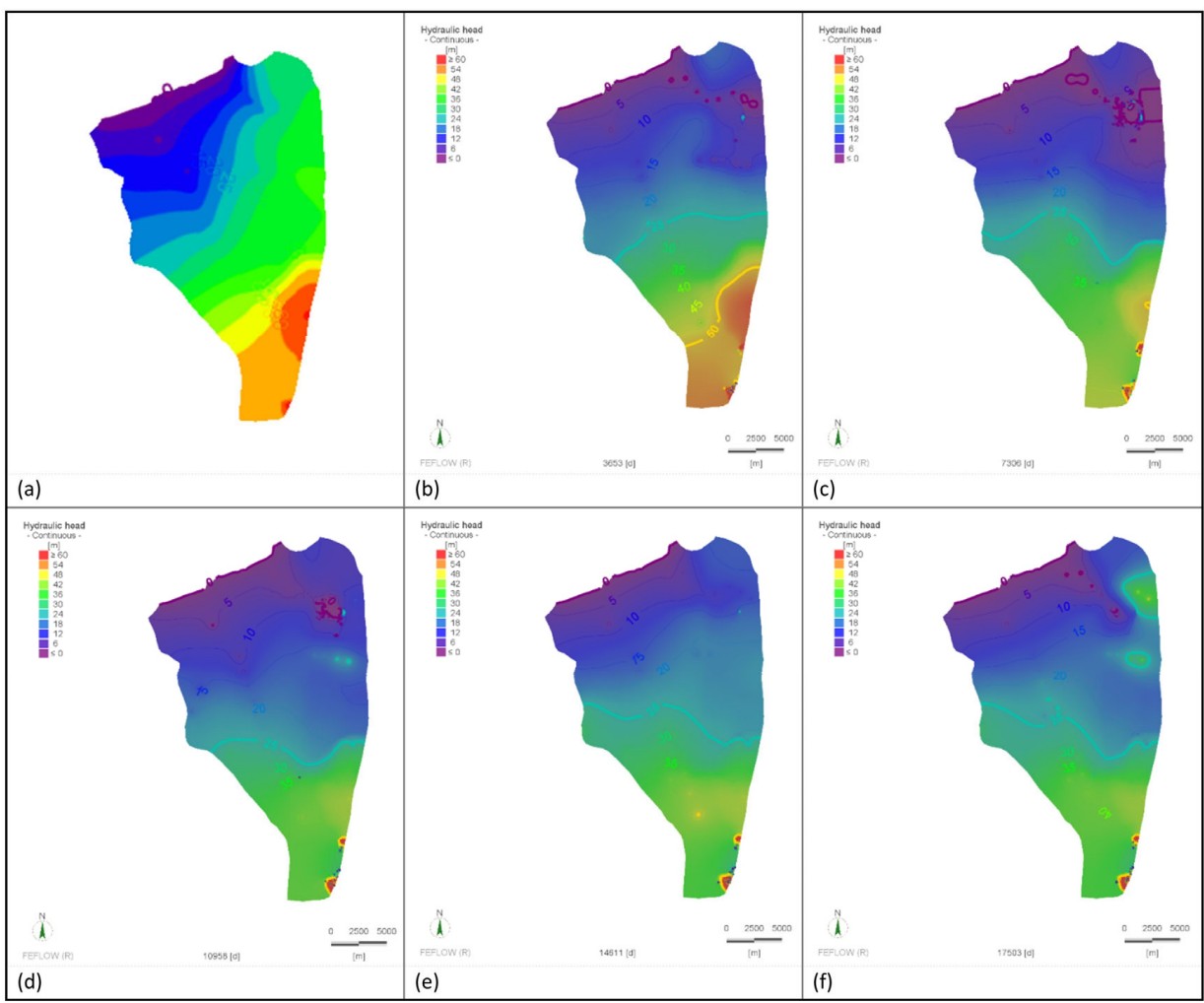

**Figure 8.** Transient flow model: starting values of the hydraulic head on 1 January 1973 (**a**), and computed hydraulic heads on 31 December 1983 (**b**), 31 December 1993 (**c**), 31 December 2003 (**d**), 31 December 2013 (**e**), and 31 December 2020 (**f**). The 25 m contour line is highlighted as a reference for the mean groundwater level.

Figure 9 shows the impact of the wadis on the global water balance of the aquifer. The cumulative (in+/out−) water volume calculated for the simulated period (1973–2020) at all the prescribed fixed nodes of the area is compared here with the water volume exchanged between the wadis and the phreatic aquifer through the 36 implemented wadi points and 12 wadi points at Oued El Bey-El Melah (Wadi Points 1, 4, 6, 8, 10, 15, 19, 23, 28, 31, 32, and 33; Figure 1d). More than three-fourths of the high water volume of 4.21 billion m$^3$ leaving the model area through the prescribed head nodes along the northern model boundary (seaside) and wadi points was due to water withdrawal from the wadis. Notably, approximately 4.02 billion m$^3$ of groundwater was withdrawn by wells during the same period. As shown in Figure 9a, the total water volume of approximately 2.65 billion m$^3$ entering the model area through the prescribed head nodes is entirely due to water infiltration from the wadis. For this reason, the time series of the cumulative water volume across the prescribed heads of the entire domain in Figure 9a is masked by the time series of the water inflow from the 36 wadi points. This further highlights that, under the given hydraulic conditions, there is no significant risk of saltwater intrusion during the simulation period. Since the calculated groundwater recharge due to precipitation corresponds to approximately 4.24 billion m$^3$, we can conclude that there is a deficit of 1.34 billion m$^3$ of water in the aquifer. This corresponds to an overuse of natural recharge of almost 50%.

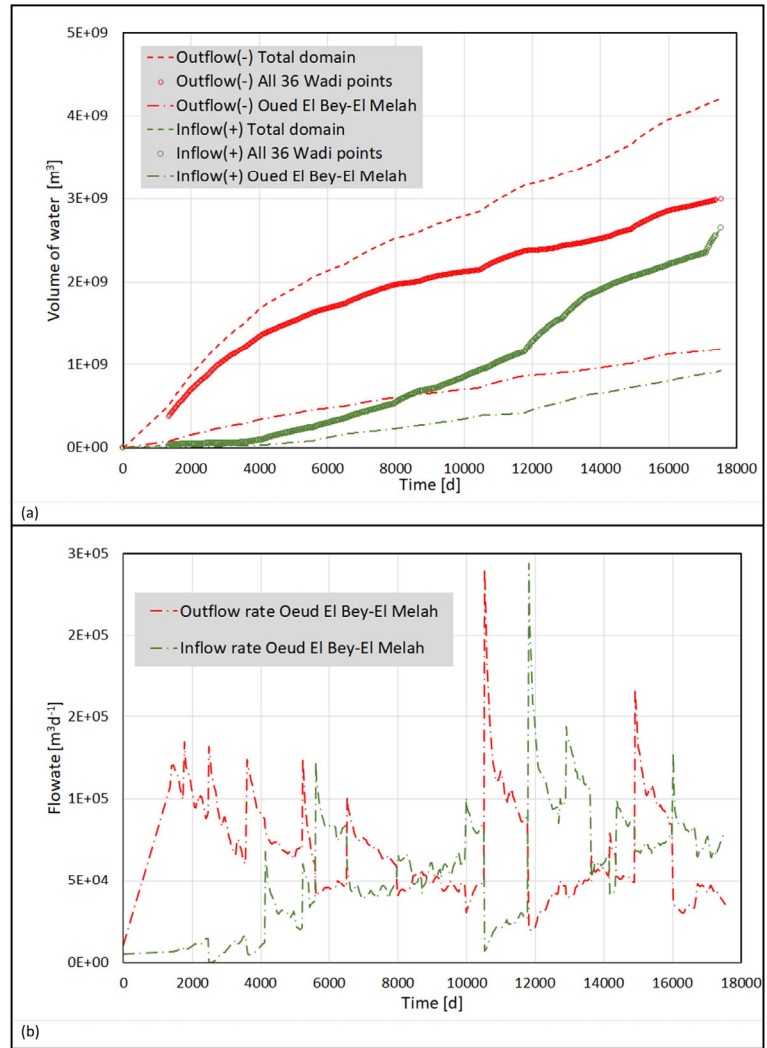

**Figure 9.** Effect of wadis on the transient water balance: (**a**) cumulative water volume computed (in+/out−) across the fixed head boundaries of the domain and across fixed head nodes located on all 36 Wadi points and those of Oued El Bey-El Melah (Wadi Points 1, 4, 6, 8, 10, 15, 19, 23, 28, 31, 32, and 33; Figure 1d); (**b**) inflow and outflow rates computed over the fixed head nodes along Oued El Bey-El Melah. The cumulative inflow water volume across the fixed head boundaries of the total domain (Figure 9a) corresponds entirely to the water volume coming from the 36 Wadi points. Note: $t = 0$ d and $t = 17{,}503$ d correspond to 1 January 1973 and 31 December 2020, respectively.

Figure 9b shows the inflow and outflow rates calculated for the prescribed head nodes along the Wadi Oued El Bey-El Melah. Most of the time, the outflow rates are much greater than the inflow rates, which explains the deficit in the cumulative water volume of approximately 0.3 billion m³ of groundwater.

The quality of the simulation is shown in Figure 10 in the form of a scatter plot of computed hydraulic heads compared to the hydraulic heads measured at 29 observation wells for 6 days: 1 January 1973; 31 December 1983; 31 December 1993; 31 December 2003; 31 December 2013; and 31 December 2020. The blue line represents the 1:1 line. It is worth noting that, in contrast to the computed hydraulic heads, some of the observed hydraulic heads sometimes had negative values, for example, 2534 on 31 December 2003 (Figure 10d) compared to 31 December 2020 (Figure 10f). For the starting date, 1 January 1973, the cross-plot of the computed hydraulic heads against the monitored hydraulic heads was qualified by a mean error ($\bar{E}$), a root mean square error (RMS), and a standard deviation ($\sigma$) of 6.57, 7.82, and 7.96 m, respectively. However, during the transient flow simulation,

these errors increased by approximately 2 to 3 m. Overall, it appears that the computed water heads were underestimated compared to the measured heads (Figure 10f). This is probably because we assumed that the annual recharge from the deeper aquifer, estimated at $7 \times 10^6$ m$^3$, remained constant during the time-dependent flow modelling. It is very likely that increasing groundwater use also increases the vertical hydraulic gradient between the underlying confined deep aquifer and the upper unconfined aquifer, leading to an increase in annual recharge from the deeper aquifer. However, the quantified deviations are quite acceptable because the ability of using databases to construct scientifically sound and reliable groundwater models is limited. It should be mentioned that data are missing on (1) monitored time-dependent water levels in the wadis, (2) observation wells at the eastern model boundary to evaluate quantitatively lateral inflow rates from the watershed, (3) detailed information on the leakage between the near-surface unconfined aquifer and the semi-deep and deep aquifers in the form of spatiotemporal hydraulic head differences and hydraulic conductance values of the separating aquifer layers, and (4) detailed pumping tests conducted to quantify spatial variation in permeability across the plane and over depth.

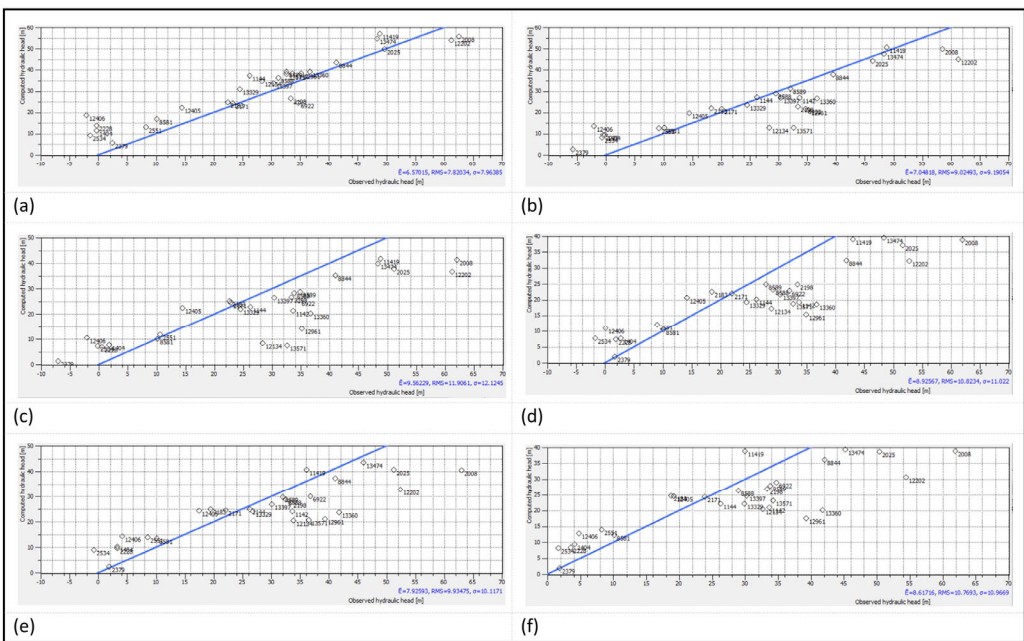

**Figure 10.** Cross-plot of the computed hydraulic heads against observed water heads at (**a**) starting values of hydraulic head on 1 January 1973 and computed hydraulic heads for 31 December 1983 (**b**); 31 December 1993 (**c**); 31 December 2003 (**d**); 31 December 2013 (**e**); and 31 December 2020 (**f**). The blue lines represent the 1:1 lines; the scatterplot is qualified by the mean error ($\overline{E}$), root mean square error (RMS), and standard deviation (σ), with all the error metrics expressed in metres.

Figure 11 shows the water heads computed at the four selected observation wells, located from north to south (Figure 1d). The four selected observation wells show typical results that were also found globally at the other measuring points.

For example, at observation well (OW) 8588 (Figure 11c), the observed time series is represented well by the numerical model. Up to 6000 days, the computed hydraulic heads were in excellent agreement with the observed values. Later, they underestimated the measured hydraulic heads by approximately 5 m but reflected the same temporal development. At OW 12,406, monitoring began in June 2000, which corresponds to a simulation period of 10,000 days (Figure 11b). In this study, the observed hydraulic heads were systematically overestimated by approximately 5 m. However, the computed hydraulic heads of OW 11,419 lied between the observed water levels (Figure 11d). At OW 2379, which is located near the sea, the computed water levels at the beginning

of the simulation (January 1973) were approximately 3 m above the observed values (Figure 11a). After 1.5 years, they fell continuously for 4000 days and reached a plateau approximately 1 m above sea level but approximately 7 m above the measured water levels. After 10,000 days, the calculated hydraulic heads were again very close to the observed values, with a difference of only a few tens of centimetres.

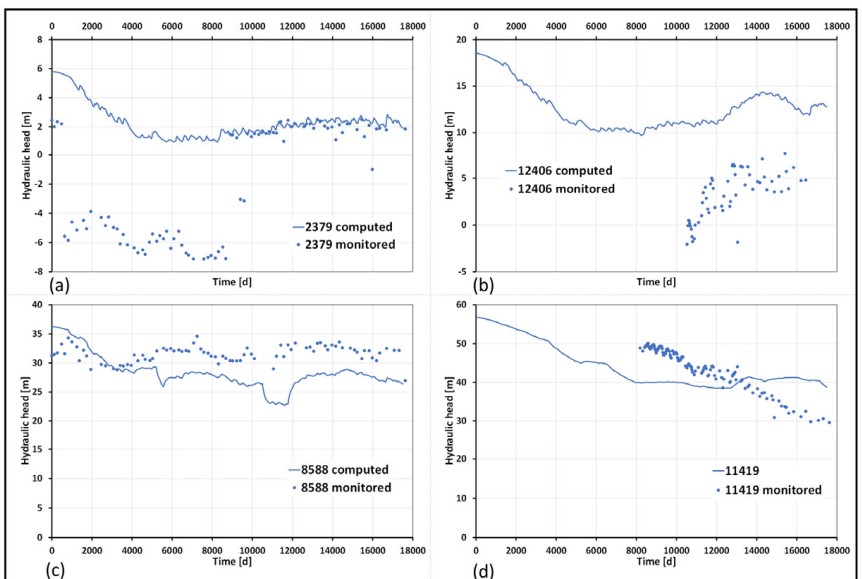

**Figure 11.** Hydraulic heads computed from 1973 to 2020 compared to those measured at the four selected observation wells, placed along a north—south longitudinal section in the study site, starting in the north: (**a**) 2379, (**b**) 12,406, (**c**) 8588, and (**d**) 11,419. The locations of the four observation wells are shown in Figure 1d. Note: $t = 0$ d and $t = 17,503$ d correspond to 1 January 1973 and 31 December 2020, respectively.

### 4.3. Modelling of Climate Change Scenarios: Groundwater Levels in the Near Future, Midterm, and Long Term

Six climate scenarios (out of the seventeen generated) were chosen as the most significant (Table 1). The six climate scenarios were based on representative concentration pathways (RCPs) 4.5 and 8.5, resulting in twelve case studies.

The scenario period was 2021–2098. To implement the chosen 12 scenarios, we used time-variant monthly averaged groundwater recharge data from eight zonal areas, deduced from predicted rainfall data from the regional climate models at 4 weather stations (Bezirk Dam, Beni Khalled Ferme Latifa, CTV Bou Argoub, Grombalia DRE) for both RCPs 4.5 and 8.5.

Table 3 shows the details of the mean annual rainfall at each of the four weather stations as a function of the period under consideration and RCP. At the Bezirk Dam weather station, which is located in the north, the mean annual rainfall predicted under RCP 4.5 is quite close to the current precipitation of 500 mm, remains almost constant from the near future to the midterm, and increases slightly in the long term compared to the near future. However, for RCP 8.5, a continuous decrease in the predicted annual rainfall from approximately 507 to 465 mm can be observed from the near future to the long-term future. In the southern region, at the CTV Bou Argoub weather station, a strong decrease in mean annual precipitation is observed for both RCPs. In the case of RCP 4.5, the mean annual precipitation decreases from 495 in the near future to 315 mm in the long term; however, for RCP 8.5, the decrease in precipitation predicted for the midterm is similar to that predicted for RCP 4.5 but lower in the long term. For the Beni Khalled Ferme Latifa and Grombalia DRE weather stations, a general trend towards a decrease in annual precipitation compared to actual precipitation is observed for both RCTPs in both the midterm and long term.

**Table 3.** Mean annual rainfall (averaged over the six climate scenarios) predicted at each of the four weather stations as a function of time and RCP considered.

| Weather Station | Representative Concentration Pathway (RCP) | Mean Annual Rainfall (mm) | | |
|---|---|---|---|---|
| | | Near Future (2021–2040) | Midterm (2041–2060) | Long Term (2081–2098) |
| Bezirk Dam | RCP 4.5 | 469 | 460 | 481 |
| | RCP 8.5 | 507 | 485 | 465 |
| Beni Khalled Ferme Latifa | RCP 4.5 | 508 | 492 | 398 |
| | RCP 8.5 | 480 | 457 | 433 |
| CTV Bou Argoub | RCP 4.5 | 495 | 448 | 315 |
| | RCP 8.5 | 512 | 458 | 379 |
| Grombalia DRE | RCP 4.5 | 598 | 560 | 464 |
| | RCP 8.5 | 493 | 478 | 416 |

Rainfall data predicted at Bezirk Dam, Beni Khalled Ferme Latifa, CTV Bou Argoub, and Grombalia DRE were applied to the zones 'RN-Perm 80 North ($PIC = 0.3$), RN-Perm 40 ($PIC = 0.6$) and RN-Perm 35 North ($PIC = 0.6$)', 'RN-Perm 80 South ($PIC = 0.3$) and RN-Perm 35 South ($PIC = 0.6$)', 'RN-Perm 25 South ($PIC = 0.15$) and RN-Perm 4 ($PIC = 0.45$)', and 'RN-Perm 25 North ($PIC = 0.15$)', respectively. Figure 12 shows the groundwater recharge applied for climate scenario 1 in both RCPs. The time series of groundwater recharge implemented in the numerical flow model varies considerably from one recharge zone to another. For example, the maximum groundwater recharge for RCP 4.5 in the 'RN Perm 35 North' zone (Figure 12d) is approximately 0.015 m per day, while the maximum value in the 'RN Perm 80 South' zone is only 0.005 m per day. The same observation can be made for RCP 8.5. In general, the time series predicted for RCP 8.5 contains globally higher values for a given zone than for RCP 4.5.

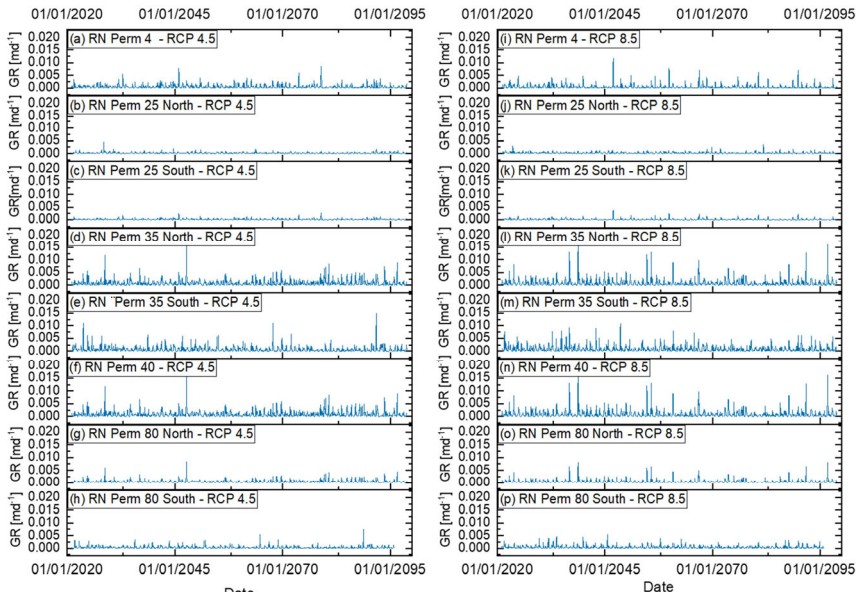

**Figure 12.** Monthly zonal groundwater recharge (*GR*) expressed in meter per day for eight distinct zones (see Figure 4a) used in climate scenario 1 based on RCP 4.5 (**a**–**h**) and RCP 8.5 (**i**–**p**). Notably, the groundwater recharges in zones 'RN35' and 'RN80' have been divided into two subzones: one located in the north and one in the south.

As in the simulation period (1973–2020), the annual recharge from the deeper aquifer, which was estimated at $7 \times 10^6$ m$^3$, remained constant during the time-dependent flow

modelling. Notably, both the pumping rate of the 740 wells and the prescribed hydraulic heads of the 36 wadi points were maintained at their 2020 values.

The hydraulic head distributions of the transient flow model, which were calculated for climate scenario 1 based on RCP 4.5, are shown in Figure 13. According to the contour maps of the hydraulic heads in Figure 13b–d, the 25 m reference contour line shifts from north to south with an increasing simulation time compared to the initial values on 1 January 2021. This illustrates the global depletion of groundwater resources predicted from a long-term perspective.

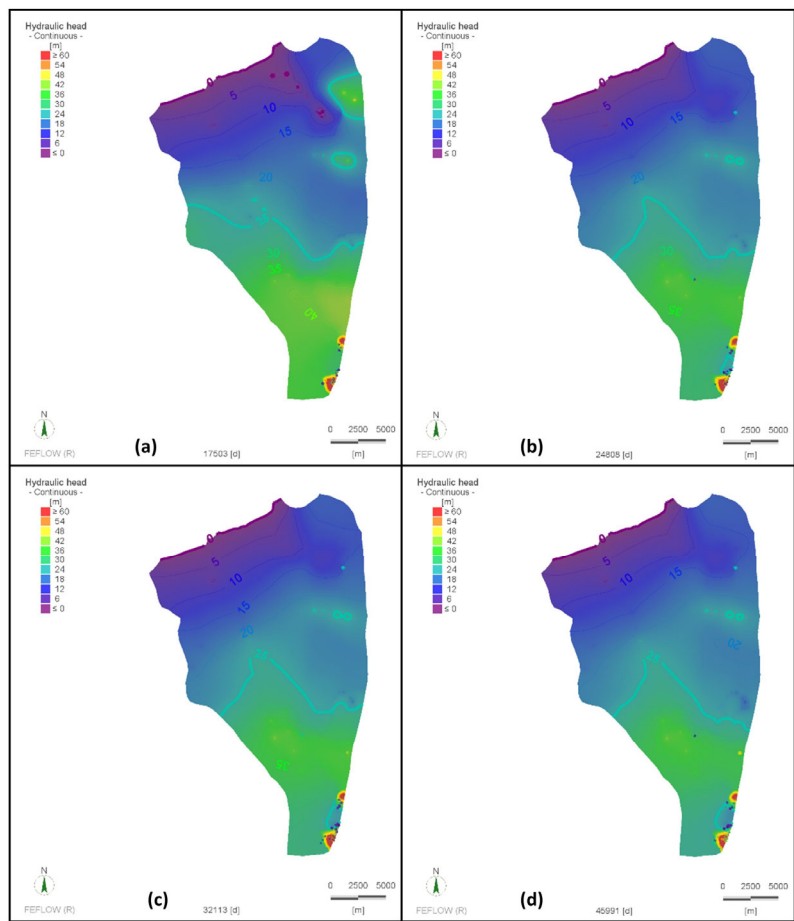

**Figure 13.** Map of the hydraulic heads computed with climate scenario 1 based on RCP 4.5 at different time stages: starting values on 1 January 2021 (**a**); 31 December 2040 (**b**); 31 December 2060 (**c**); 31 December 2098 (**d**). The 25 m contour line is highlighted as a reference for the mean groundwater level.

To illustrate the effects of climate scenario 1 on the development of the groundwater level in more detail, the results for the four selected observation wells are shown as examples in Figure 14. The observation wells are arranged along a north—south longitudinal section in the study area (see Figure 1d), starting in the north: 2379, 12,406, 8588, and 11,419. In the case of RCP 4.5, a general trend of decreasing hydraulic heads with increasing time is observed for observation wells 11,419 and 8588. The largest decrease is predicted for the southernmost observation well, observation well 11,419. However, for observation wells 12,406 and 2379, which are both located further north than the previous two wells, slight increases in the hydraulic heads are observed compared to the reference head in January 2021. While the same observations can be made for observation wells 11,419 and 8588 in the case of RCP 8.5, the long-term predicted hydraulic heads for observation wells 12,406 and 2379 are characterized by a general trend of a very slight decreases in hydraulic heads.

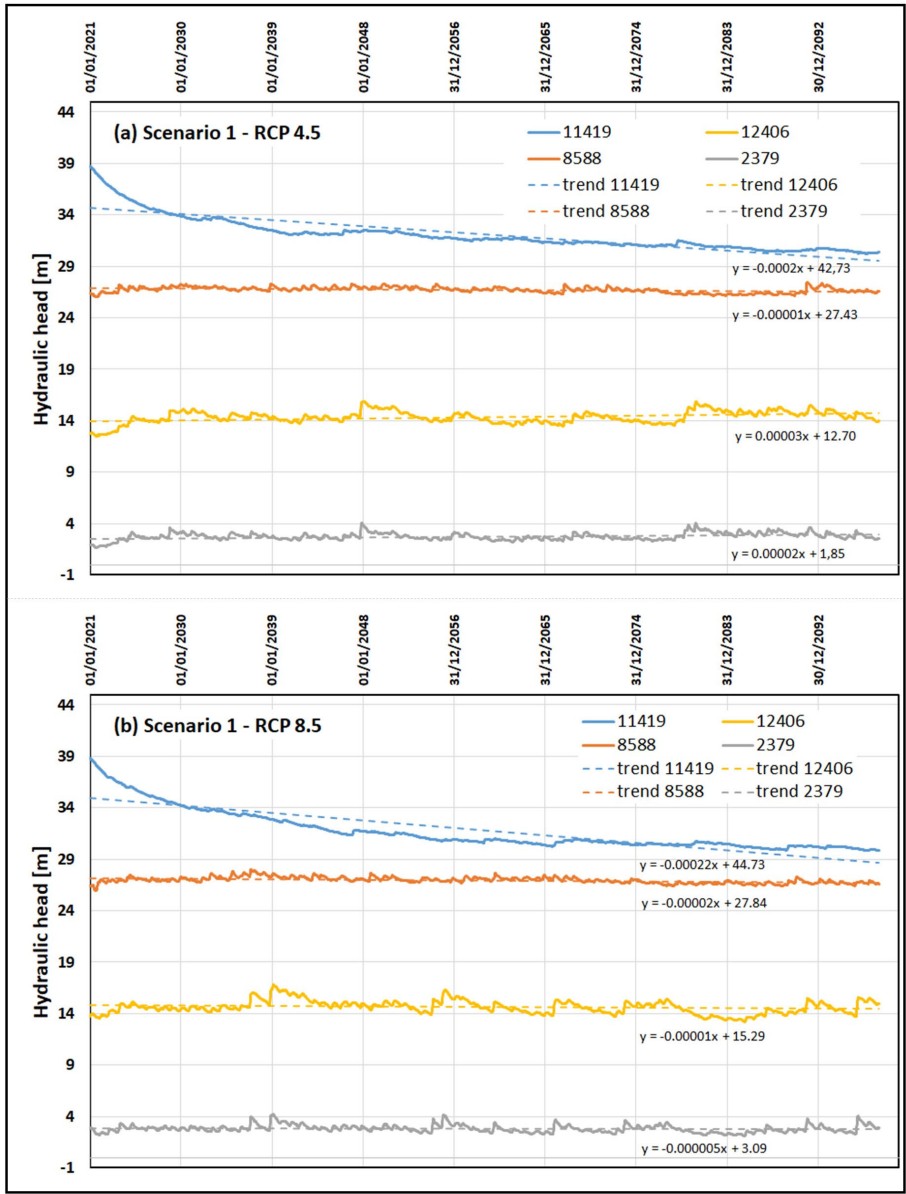

**Figure 14.** Results of the time-dependent hydraulic head calculations (from January 2021 to December 2098) under climate scenario 1 for RCP 4.5 (**a**) and RCP 8.5 (**b**) at the four selected observation wells.

More detailed information on the temporal variation in the hydraulic heads computed at the individual observation points can be found in the Supplementary Materials. Tables S2 and S3 show a summary of the three times and all 34 observation wells as well as the time averages of the minimum, mean, and maximum values of hydraulic heads based on the six climate scenarios for RCP 4.5 and RCP 8.5, respectively. The three times considered are the near future (2021–2040), the midterm perspective (2041–2060), and the long term perspective (2081–2098).

Figure 15 shows the histogram of the variation in hydraulic heads at the 34 observation wells, averaged over six climate scenarios, compared to the baseline hydraulic heads in 2021. The figure illustrates the number of observation wells that fall within the same range of hydraulic head variation, divided into equal bins of 2 metres. The histogram summarizes an interesting result with regard to the long-term perspective (2081–2098): between RCP 4.5 and RCP 8.5, only small differences in the predicted fluctuations in the hydraulic head can be observed. For example, in the case of RCP 4.5, increases of up to 2 m in the hydraulic head are predicted for 15 observation wells, while 13 observation wells are counted for

RCP 8.5. A decrease in the hydraulic head of more than 6 m was observed in 4 observation wells located in the southern region for both RCPs. In general, the results of the climate scenarios reveal a bi-structured north—south behaviour in the hydraulic heads: an increase in the north and a decrease in the south. In the southern part of the study site, due to the reduced annual rainfall (−20%) predicted by the various scenarios (see Table 3), a significant decrease in the groundwater level of up to 10 m was observed between 2021 and 2098. In the northern part, the predicted rainfall is the same as that today, or even slightly increased. Therefore, hydraulic heads are either increased or stable. Notably, a further increase in the pumping rate due to climate change would be severe for the southern part of the Wadi El Bey aquifer.

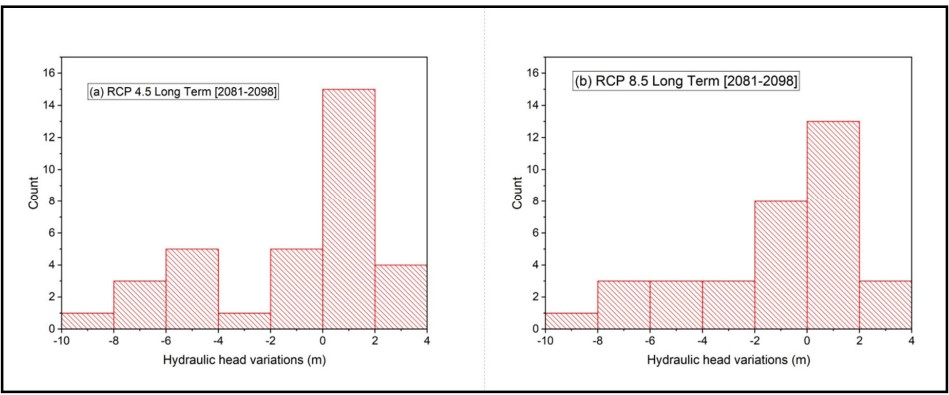

**Figure 15.** Histogram of the hydraulic head differences calculated for RCPs 4.5 (**a**) and 8.5 (**b**) from a long-term perspective with regard to baseline hydraulic heads in 2021.

## 5. Conclusions

The numerical 3D groundwater model developed in this study contributes to improving the modelling of transient groundwater flow in the coastal area of the Wadi El Bey plain. Furthermore, our modelling approach included climate change scenarios to simulate the influence of future rainfall data on predicted groundwater levels, which may significantly impact both the available groundwater quantity and groundwater quality. To be effective and close to field operations, we used the calibrated numerical tool FEFLOW to model the groundwater flow. Our modelling approach was based on a three-dimensional spatial discretization of the physical domain of the aquifer.

Assessing anthropogenic and natural effects on groundwater resources in the Wadi El Bey coastal aquifer was particularly challenging because of the lack of data on surface water and groundwater recharge. Using GIS model-based information, we were able to implement prescribed water head nodes spatially distributed along the main wadis of the study area, which helped to reduce the uncertainty of potential water exchange between surface water and groundwater in the spatial data.

The methodology presented in this paper is based on a novel modelling approach that considers zonal groundwater recharge when estimating transient groundwater flow in coastal areas. In the flow simulations, we evaluated effective vertical transient groundwater recharge using the local potential infiltration coefficient, which is a function of the soil surface texture and monthly precipitation. Compared to the full hydrologic water balance approach used in other field studies, our modelling approach is much simpler for practical application and can be easily implemented in existing groundwater flow models.

In combination with spatially distributed recharge calculations through hydrological modelling, the sparsely available data were enriched, allowing for the reconstruction of general trends. The cumulative (in and outflowing) water volume computed for the simulated period (1973–2020) indicates that there was a deficit of 1.34 billion m$^3$ of water in the aquifer. This corresponds to an overuse of natural recharge of almost 50%. The importance of water exchange between wadis and groundwater and its impact on the

overall water balance should be emphasized. Water exchange also has a direct impact on groundwater quality, as surface water is increasingly affected by wastewater discharges of heavy metals, pollutants, and salts from industrial and urban activities. The simulations showed that more than three-quarters of the high water volume of 4.21 billion $m^3$ leaving the model area via the prescribed head nodes along the northern model boundary (sea side) and the wadi points was due to water withdrawal from the wadis. The total water volume of approximately 2.65 billion $m^3$ entering the model area through the prescribed head nodes was entirely due to water infiltration from the wadis. This further highlighted that under the given hydraulic conditions, there is no significant risk of saltwater intrusion during the simulation period.

The quantified deviations in the computed hydraulic heads from the measured water levels are quite acceptable because the database used to construct a scientifically sound and reliable groundwater model was limited. Further work is required to collect field data to quantitatively assess local inflow/outflow rates between surface water and groundwater. The simulation of 12 climate scenarios highlighted a bi-structured north—south behaviour in hydraulic heads: an increase in the north and a depletion in the south. A further increase in the pumping rate would, thus, be severe for the southern part of the Wadi El Bey aquifer.

Even in the absence of sufficient data, the presented modelling framework enables large-scale assessment of complex aquifers. The realistic results produced may help resource managers evaluate their management strategies. Although it was developed for the Wadi El Bey aquifer, this method can be applied to other areas around the world. In particular, it may be possible to estimate groundwater reservoir development and investigate the long-term effects of variations in groundwater recharge caused by climate change.

**Supplementary Materials:** The following supporting information can be downloaded at: https://www.mdpi.com/article/10.3390/w16040522/s1, Figure S1: Monthly rainfall predicted at the Bezirk Dam weather station for each of the six selected scenarios for (a) RCP 4.5 and (b) RCP 8.5; Figure S2: Monthly rainfall predicted at the Ben Khalled Ferme Latifa weather station for each of the six selected scenarios for (a) RCP 4.5 and (b) RCP 8.5; Figure S3: Monthly rainfall predicted at the CTV Bou Argoub weather station for each of the six selected scenarios for (a) RCP 4.5 and (b) RCP 8.5; Figure S4: Monthly rainfall predicted at the Grombalia DRE weather station for each of the six selected scenarios for (a) RCP 4.5 and (b) RCP 8.5; Table S1: Coordinates of the DGRE observation wells, artificial recharge basins, virtual observation points and wadi points. All virtual observation points and wadi points are located in the first (upper) slice of the numerical groundwater model; Table S2: Results of scenarios based on RCP 4.5: time averages of the minimum, mean and maximum values of computed hydraulic heads. OW = observation well; Table S3: Results of scenarios based on RCP 8.5: time averages of the minimum, mean and maximum values of the computed hydraulic heads. OW = observation well.

**Author Contributions:** Conceptualization, H.B., M.L. and G.S.; data curation, Y.A. and G.S.; modelling, H.B., M.L. and G.S.; formal analysis, H.B., M.L. and G.S.; funding acquisition, H.A., T.M. and G.S.; methodology, G.S.; project administration, H.A., T.M. and G.S.; supervision, H.A., T.M. and G.S.; visualization, H.B., Y.A. and G.S.; writing—original draft, G.S.; writing—review and editing, H.B., M.L., H.A., T.M., Y.A. and G.S. All authors have read and agreed to the published version of the manuscript.

**Funding:** The research study was funded by the General Secretariat for Research and Technology of the Ministry of Development and Investments under the PRIMA Programme. PRIMA is an Art. 185 initiative supported and co-funded under Horizon 2020, the European Union's Programme for Research and Innovation. We also acknowledge funding from the Ministry of Higher Education and Scientific Research of Tunisia.

**Data Availability Statement:** The data presented in this study are available on request from the corresponding author.

**Conflicts of Interest:** On behalf of all the authors, the corresponding author states that there are no conflicts of interest. The funders had no role in the design of the study; in the collection, analyses, or interpretation of data; in the writing of the manuscript; or in the decision to publish the results.

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
