# Peer review of "Quantitative Groundwater Modelling under Data Scarcity: The Example of the Wadi El Bey Coastal Aquifer (Tunisia)"

_water, doi:10.3390/w16040522_

Round 1

Reviewer 1 Report

Comments and Suggestions for Authors

Water 2023, 15: “Quantitative groundwater modelling under data scarcity: the example of the Wadi El Bey coastal aquifer (Tunisia)”.

Open Review

Comments and Suggestions for Authors.

The paper deals with an interesting topic. The purpose of this study is to investigate shallow groundwater level’s changes to long-tern perspective in coastal aquifer in Tunisia using a numerical computer modelling approach.   

Facts and analysis of procedures confirm its great potential. 

Nevertheless, it needs definitely to be rewritten before publication. The language needs to be checked again and preferably by a native speaker.

The following points need to be covered:

1.     There is no clear explanation of what data is scarcity according to the title of the manuscript.

2.     The introduction should clearly state the objectives of the study.

3.     The introduction (lines 109-126) presents the results of this study. They should be shown in the conclusion, or in the methods and/or in results.

4.     Various geographical names that the authors refer to during the analysis are not indicated on any map (diagram), for example: Bou Argoub, Mejerda-Cap Bon canal, Bezirk Dam, El Gobba, etc.

5.     The article is overloaded with numerous abbreviations that make it difficult to understand the meaning. For example, table 1 with a description of climate model scenarios.

6.     The results of calibration of the transient flow model raise doubts both in terms of the magnitude of the discrepancies between the observed and calculated model values, and in terms of the correspondence to the dynamics of fluctuations in groundwater levels (Figure 11).

7.     The inscriptions on the pictures are unreadable.

Specific comments and suggestions.

Introduction. Lines 49-51. The text requires editing, since irrigation, as a rule, increases groundwater levels, and its reduction - decreases it.

Line 166. Surface wells? Maybe it is shallow wells.

Lines 214-215. “The constructed 3D numerical model is composed 214 of three layers (and four slices)….” What are layers and slices represented? It must be explained.

Reviewer 2 Report

Comments and Suggestions for Authors

Thank you for the opportunity to review your manuscript.  I found the level of detail of modeling quite extensive, but have many questions and request some clarifications.  My main concern is the poor transient model agreement with observed data.  Could the model be simplified to improve prediction?  Short of collecting additional data, what modeling approaches could be used to improve prediction of observed values?  Perhaps some discussion of this could be included.  Finally, I am concerned about the validity of the climate change model results given the poor agreement of the transient model.  Perhaps you could include discussion of why you believe the model results are robust enough to predict future trends.

Comments on the Quality of English Language

English is ok

Reviewer 3 Report

Comments and Suggestions for Authors

The manuscript by Baccouche et al. is an interesting case study using numerical modelling techniques of a coastal aquifer threatened by over-pumping.

The manuscript is well written. I have only few comments for this interesting study.

L 33: “Coastal aquifers are therefore extensively exploited, and saltwater intrusion has been observed in many areas [3]. Additional could be given here on this topic as well as in the discussion section.

L 49 “that the level of the water table” Simplify to just “water table”

L163: “The flow direction of groundwater is globally directed towards the north” Should be”generally” instead of “globally”.

Figure 1b) The legend of the cross section is too small and cannot be read easily.

Figure 1c) It is not clear where the artificial recharge basins are in the figure. Are these the yellow lines?

L 228: “In the steady-state flow model, the prescribed hydraulic heads of the chosen wadi points were extracted from the study of Gaaloul et al. [13].” It is understood that the hydraulic heads were taken from a different study. How was the level of these hydraulic heads determined in the first place?

L231: “The hydraulic heads were calculated using a triangular interpolation technique from 1972 to 2019 232 [25] from which hydraulic heads were monitored at observation wells in April and 233 October.” This sentence needs to be rephrased.

“The GR was calculated for each areal zone zone i considering the local values of both R and PIC for each geological formation in the model domain [27] “. Reference 27 is not an easily accessible reference as it is a report. Is there any other reference that can be cited if readers want to follow this approach?

Figure 3: The zonation shown in this figure, is it chosen based on lithology or other properties?

Figure 4: The resolution of this figure as well as other figures can be improved.

L 385 “the greatest amounts of artificial recharge occurred” Is this a modelling result or was this measured? Was the artificial recharge varied during the transient simulations?

Figure 10: The copy and past screenshot figures resolution is not very good. This figure can be improved.

Figure 10: Some of the RMS errors are quite high. What could be the reason for the higher RMS errors during the transient simulation?

L 541L “The water inflow rate from the deeper aquifer was maintained as in the transient flow model for the simulation period (1973-2020).” This sentence needs rephrasing.
